# Vaccine-Based Immunotherapy for Head and Neck Cancers

**DOI:** 10.3390/cancers13236041

**Published:** 2021-11-30

**Authors:** Simon Beyaert, Jean-Pascal Machiels, Sandra Schmitz

**Affiliations:** 1Institut de Recherche Expérimentale et Clinique (IREC), Pôle MIRO, Université Catholique de Louvain (UCLouvain), 1200 Brussels, Belgium; 2Department of Medical Oncology, King Albert II Cancer Institute, St-Luc University Hospital, 1200 Brussels, Belgium; jean-pascal.machiels@uclouvain.be; 3Department of Head & Neck Surgery, King Albert II Cancer Institute, St-Luc University Hospital, 1200 Brussels, Belgium; sandra.schmitz@uclouvain.be

**Keywords:** vaccine, immunotherapy, head and neck, squamous cell carcinoma, undifferentiated nasopharyngeal carcinoma, human papillomavirus, Epstein–Barr virus

## Abstract

**Simple Summary:**

Therapeutic vaccines are given to patients with cancer, as opposed to prophylactic vaccines given to a healthy population. The challenge for therapeutic oncological vaccines is to stimulate an immune T cell response against endogenous (or derived) antigens that is sufficiently potent to induce cytotoxic activity and broad enough to take tumor heterogeneity into account. The purpose of this article is to provide an updated review of the prophylactic and therapeutic vaccines that target viral or non-viral antigens, particularly in head and neck cancers.

**Abstract:**

In 2019, the FDA approved pembrolizumab, a monoclonal antibody targeting PD-1, for the first-line treatment of recurrent or metastatic head and neck cancers, despite only a limited number of patients benefiting from the treatment. Promising effects of therapeutic vaccination led the FDA to approve the use of the first therapeutic vaccine in prostate cancer in 2010. Research in the field of therapeutic vaccination, including possible synergistic effects with anti-PD(L)1 treatments, is evolving each year, and many vaccines are in pre-clinical and clinical studies. The aim of this review article is to discuss vaccines as a new therapeutic strategy, particularly in the field of head and neck cancers. Different vaccination technologies are discussed, as well as the results of the first clinical trials in HPV-positive, HPV-negative, and EBV-induced head and neck cancers.

## 1. Introduction

Immune checkpoint inhibitors and targeted therapies are a major oncological breakthrough and are currently approved by the U.S. Food and Drug Administration (FDA) as treatment options for several cancers. Following the results of the phase III Keynote-048 study [1], pembrolizumab, a monoclonal antibody targeting PD-1, was approved in 2019 for the first-line treatment of recurrent and/or metastatic (R/M) head and neck squamous cell carcinoma (HNSCC) [2]. Despite the promising results of immune checkpoint inhibitors (ICI) such as pembrolizumab compared to other treatment strategies in R/M HNSCC, response rates of only 13–20% have been documented [3,4]. Therefore, tools to predict response and resistance to anti-programmed death (PD)-1 therapies would be useful. 

Patients with “hot tumors” seem to currently have a more favorable outcome. These tumors are defined as being in an inflammatory microenvironment with a high immune gene expression signature, high level of tumor-invasive T-lymphocytes (TIL), a strong tumor expression of PD-ligand(L)1, and a high mutational or neoantigen burden. Several studies have highlighted that HNSCC with a high level of TIL invasion, or high expression of immune-related genes, has a better overall survival, disease-specific survival, and disease-free survival [5,6,7]. These non-independent markers differ from “cold tumors” that respond poorly to ICIs and are associated with poor survival [8]. To increase efficacy, prognostic tools and new immunomodulatory strategies or treatment combinations are of interest. 

In cancer cells, many neoantigens are produced and expressed from non-synonymous gene mutations. Their non-expression in normal tissues and their strong immunogenicity make these neoantigens ideal targets for immunotherapy [9]. Due to the interactions between the immune system and cancer cells, spontaneous T cell responses have been detected against these antigens. However, in most patients, this anti-cancer T cell response is rendered inactive by the tumor and its immunosuppressive microenvironment [10,11]. The discovery of these neoantigens has therefore revealed immune responses directed against these tumor antigens, giving way to research into vaccine-based therapies.

Therapeutic vaccines are given to patients with cancer, as opposed to prophylactic vaccines given to a healthy population. Therapeutic vaccines face the challenge of stimulating and strengthening the patient’s adaptive immune system by increasing the anti-tumor immune reaction to attack cancer cells. The aim is to spare healthy cells in order to provide clinical effects and/or to improve the clinical response in combination with other therapies [3,12]. The purpose of this article is to provide an updated review on vaccine-based therapies, particularly in head and neck cancers.

### 1.1. Therapeutic Vaccines 

Therapeutic vaccines are currently under investigation for established oncological disease. With the aim of stopping tumor growth, therapeutic vaccines should be investigated in combination with other management strategies, such as immunotherapies, to see if their effects can be potentiated. In 2010, the FDA approved the use of sipuleucel-T, the first cell-based therapeutic vaccine to demonstrate a survival benefit in patients with hormone-resistant prostate cancer. Despite encouraging progress, significant effort is still required to enhance the effectiveness of vaccine-based treatments [12]. 

Traditional vaccines against infectious diseases are highly effective because they induce a humoral reaction and the production of immunoglobulins directed against exogenous epitopes. Oncological therapeutic vaccines as single agents can also induce a humoral reaction, but it is insufficient to control cancer and results in only rare objective clinical benefits. The challenge of therapeutic oncology vaccines is therefore to stimulate an immune T cell response against endogenous (or derived) antigens that is sufficiently potent to induce cytotoxic activity and broad enough to take tumor heterogeneity into account [13]. There are two main classes of tumor-associated antigens (TAAs): antigens with high tumor specificity and antigens with low tumor specificity. Antigens with high tumor specificity include another three types of antigens: virus-related antigens (such as antigens from HPV16 E7/E7 proteins), antigens related to non-synonymous mutations in genes producing neo-antigens, and antigens encoded by cancer germline genes (such as the MAGE gene family). Antigens produced by cancer germline genes are normally expressed only in germline cells. In the periphery, these antigens can be presented by cancer cells and can generate immune reactions without creating a T-immune reaction against the germ cells due to their lack of major histocompatibility complex (MHC) I expression. On the other hand, antigens with low tumor specificity include differentiation antigens (e.g., CEA), which are found only in tumor cells and in the original normal tissue, and antigens produced by overexpressed proteins (e.g., p16). Indeed, some over-expressed non-mutated proteins can induce a T cellular immune response if the amount of HLA-peptide complexes exceeds a certain threshold compared to healthy cells [14]. 

An additional interesting mechanism is also the concept of “epitope spreading”. This principle is based on the fact that a primary immune reaction with a well-defined target (such as a tumor antigen for an oncological therapeutic vaccination or via the blocking of an immune checkpoint) leads to tumor cell death. These dead tumor cells will be phagocyted by antigen-presenting cells that will then present other tumor antigens on their MHC I and II and stimulate new naïve T cells directed against these other tumor antigens [15]. In a pre-clinical study, carcinoembryonic antigen (CEA)-transgenic mice transplanted with CEA(+) tumors were vaccinated with CEA/TRICOM, a poxviral vector vaccine that contains the transgenes for CEA and a triad of T cell co-stimulatory molecules. A specific immune response to CEA was demonstrated, as was a response against additional antigens expressed on the tumor itself as wild-type p53 and an endogenous retroviral epitope of gp70. Surprisingly, the majority of T cells infiltrating the regressing CEA-positive tumor were specific for gp70 [16].

### 1.2. Main Therapeutic Vaccine Strategies

#### 1.2.1. Peptide-Based Vaccines

Cancer cells produce peptide antigens that are presented on their membrane surface via the MHC. These antigens can be recognized by the T cell receptor (TCR) of cytotoxic T lymphocytes (CTL), resulting in cancer cell lysis (Figure 1) [17]. 

Peptide-based vaccines are composed of amino acid sequences containing the epitope that can trigger an immune response. The injected peptide is taken up by antigen-presenting cells that then present to naïve T cells in a human leukocyte antigen (HLA) pathway. The activated cytotoxic T lymphocytes will then recognize the same epitope presented on the surface of the tumor cells via MHC I in order to eliminate the tumor cells [18]. 

There are two types of peptide sequences currently being used in therapeutic vaccines: short peptides (SPs) (consisting of about 10 amino acids) and long peptides (LPs) (consisting of between 25–35 amino acids) [17,19]. LPs have several advantages over SPs. SPs have the ability to bind directly to MHC I molecules, i.e., some non-professional antigen-presenting cells (such as fibroblasts) are involved in the immune process without optimized co-stimulation, which is less effective than if the T cells are stimulated by professional antigen-presenting cells such as a dendritic cells (DCs). This disadvantage can be avoided by using a LP vaccine, forcing phagocytosis of the LP by DCs before the epitope is exposed on MHC I to be presented to T cells. LP vaccines also broaden the HLA-related compatibility that may exist with a SP vaccine. Moreover, the use of a LP also allows presentation via MHC II of antigen-presenting cells, and thus the stimulation of CD4+ lymphocytes, allowing a more effective immune response against tumor cells. Th1 stimulation also allows the production of inflammatory cytokines (e.g., interferon (IFN)-γ, tumor necrosis factor (TNF)-α, interleukin (IL)-2) that may aid immune responses against the tumor [17,20,21], promote the formation of memory T-lymphocytes that play an important role in the long-term maintenance of the immune response [17,22], and participate in cancer cell destruction through the stimulation of macrophages [23]. However, peptides are not very immunogenic on their own, so concomitant administration of an adjuvant is essential for effectiveness [12]. This type of vaccine is known to be safe, as demonstrated in many clinical trials [13,24,25].

#### 1.2.2. DC-Based Vaccines

As described above, DCs are key to initiating an effective adaptive immune response. DCs search for antigens in peripheral tissues. When an antigen is recognized and internalized, the activated DCs migrate to the draining lymph node to induce an adaptive immune response through naïve T cells. The presentation of internalized antigens on MHC I molecules is a process called cross-presentation, and it is composed of two main pathways: the vacuolar pathway and the endosome–cytosol pathway. In the vacuolar pathway, antigen presentation onto MHC I molecules happens in the endo/lysosomal compartment where antigens are decomposed by lysosomal proteases (such as cathepsin S), and where peptides derived from this degradation are loaded onto MHC I molecules. In the endosome–cytosol pathway, internalized antigens must be transported to the proteasome. The resulting peptides are then transported by the transporter associated with antigen processing (TAP) into the endoplasmic reticulum or the antigen-containing endosomes, where they can be loaded onto MHC class I [26]. In the MHC II-restricted presentation, internalized antigens are decomposed by endo/lysosomal proteases, such as cathepsins. Newly synthesized MHC II molecules, which are stabilized by the attachment to the invariant chain (Ii), are transported from the endoplasmic reticulum to this endo/lysosomal compartment where Ii is then degraded by proteases, resulting in the binding of a small peptide fragment (CLIP) to MHC II. Thereafter, CLIP is replaced by peptides derived from the antigen of interest by the HLA-DM chaperone [26].

For DC-based vaccines, DCs are isolated from autologous peripheral blood mononuclear cells and then matured ex vivo until they express MHC class I and II. They are then loaded with relevant tumor antigens before being injected into the patient to induce immune stimulation to these tumor antigens [27]. Local reactions are common and are in contrast to systemic grade 3–4 reactions that are extremely rare in monotherapy [28]. Dendritic vaccination has already been investigated in several clinical trials. The results of a meta-analysis indicate that approximately 77% of prostate cancer patients and 61% of renal cell carcinoma patients present an antigen-specific cellular immune response to dendritic cell vaccines [29]. Despite good safety and evidence of immunogenicity, DC vaccines are considered to have low therapeutic efficacy. Several factors may provide an answer to explain these poor clinical results. First, tumor cells can inhibit MHC I expression on their surface, making them unidentifiable by effector T cells. In addition, tumor cells may express inhibitory immune checkpoints, such as CTLA4 and PD-L1, which may inhibit the cytolytic effect of effector T cells activated by DCs, providing a rationale for the use of combination therapy. In addition, the expression of immunomodulating enzymes, such as indoleamine-2,3-dioxygenase 1 (IDO-1) by cancer cells and other peri-tumoral cells, can alter host DC differentiation, maturation, and functionality, and inhibit the activation of effector T-cells, promote the transformation of naïve T cells into regulatory T cells (Tregs), and induce a peritumoral immunotolerant environment towards cancer cells [30,31]. Another potential factor is that studies investigating DC-based vaccines have been performed in patients with multiresistant cancers who have already benefited from several lines of treatment, and this may underestimate the biological activity and efficacy of this type of treatment. The best example of efficacy with this type of vaccine is observed in the IMPACT study. This randomized phase III trial tested the effectiveness of sipuleucel-T, an antigen-presenting cells (APCs)-based therapeutic vaccine, in hormone-resistant metastatic prostate cancer. The investigators stimulated autologous APCs ex vivo with the recombinant fusion protein PA2024, which consists of a prostate antigen and prostatic acid phosphatase combined to granulocyte-macrophage colony-stimulating factor (GM-CSF). The vaccine was found to improve median overall survival (OS) compared to placebo (25.8 months versus 21.7 months, respectively) [28,32]. Although less than five percent of patients had an objective response rate, sipuleucel-T was approved by the FDA in 2010 [12,28,33]. These encouraging results demonstrate the need to improve and optimize this therapeutic approach.

#### 1.2.3. DNA Vaccines

DNA vaccines are made from bacterial plasmids encoding one or more tumor antigens with possibly other inflammatory molecules. Once in the nucleus, the bacterial plasmid is expressed in order to deliver its tumor antigens, which are then presented to T cells via MHC I and II to trigger the activation of CD8+ and CD4+ T cells. Again, the aim of these vaccines is to stimulate the adaptive immune system against tumor antigens. In addition, bacterial plasmids naturally create an innate inflammatory reaction due to the presence of CpG motifs and the double stranded structure itself [12,34] through toll-like receptors (TLRs) and via multiple cytosolic DNA sensors such as absent in melanoma (AIM)2, interferon-γ-inducible protein (IFI)16, cGAMP synthase (cGAS), STING, and others [35]. An advantage of plasmid vaccines is their ability to integrate many genes encoding multiple tumor-antigens (also called poly-epitope DNA vaccine), to create a precise and broader adaptive immune response at the same time. This also compensates for the loss of vaccine efficacy if tumor cells mutate or delete antigens, or if the patient does not have the appropriate T cell repertoires or MHC incompatibility.

However, DNA vaccines are poorly immunogenic. To increase the immune response to these vaccines, several strategies have been used: the inclusion of cytokine coding genes in plasmids (e.g., IL-2, IL-12, GM-CSF, INF-γ, etc.), the use of chimeric DNA vaccines, adjuvants, or a combination of all. To date, DNA vaccines are still in phase 1 or 2 studies. According to data from completed studies, DNA vaccines are considered safe [21,34]. 

#### 1.2.4. RNA Vaccines

The idea of creating RNA vaccines emerged at the same time as the thought around DNA vaccines. However, DNA vaccines were initially favored because RNA was known to be expensive, difficult to manufacture, and unstable. RNA vaccines have, however, become much more attractive since their initial concept.

RNA strands are injected after being combined with a lipid carrier, allowing both their protection in the extra-cellular environment and their absorption by DCs. Exogenous RNA is inherently immunogenic because it is detected by innate immune receptors present on the cell surface—in the cytosol and in endosomal vesicles (as toll-like receptors (TLRs)). Other methods are also used to increase the immunogenicity of RNA-based vaccines, such as the injection of adjuvant, or the encoding of immunostimulatory proteins in the RNA strand (as CD70, CD40L, and constitutively active TLR4) [36,37].

RNA strands can be expressed in the cytosol of cells and, unlike DNA vaccines, do not need to be integrated into the nucleus. This avoids the accidental insertion of genetic mutations or permanent genomic alterations [12,36,38,39]. After RNA translation, the encoded proteins are transformed into peptides that present on MHC I and II to stimulate CD8+ and CD4+ T cells, respectively. RNA vaccines have been shown to be safe. Like DNA vaccines, RNA vaccines are less dependent on the patient’s genetic background in the presence of encoded peptides as opposed to SP vaccines [36,39].

Another vaccine strategy involving RNA is a mRNA-transfected DC vaccine [40,41,42], in view of the central role of DCs in triggering adaptive immunity by producing CTL in in vitro and in vivo situations. 

The most commonly used strategy for maturing DCs is the use of synthetic peptides derived from tumor antigens. However, peptides are dependent on the primary identification of a patient’s HLA status, meaning that only some patients with specific HLA can be treated with this modality. Although this problem has been overcome with the use of LP, RNA molecules represent an alternative to the stimulation of DCs regardless of the patient’s genetic background. Furthermore, since the protein is expressed after RNA transcription in cells such as DCs, several peptide-MHC I and II complexes are created [41].

#### 1.2.5. Live Vector-Based Vaccines

Vaccines based on live vectors use bacteria or viruses. Several bacteria, including *Listeria monocytogenes*, *Lactobacillus lactis, Lactobacillus plantarum,* and others, have been explored to produce therapeutic cancer vaccines [38,43]. For viruses, adenovirus, vaccinia virus, alpha-virus, and others have also been investigated.

*Listeria monocytogenes* is a promising vector because this bacterium triggers an innate and adaptive immune response by infecting macrophages and dendritic cells, and secreting a toxin called listeriolysin O (LLO) to evade phagosomal lysis. These live vaccines can replicate within host cells as DCs and express tumor antigens on MHC I and II. They also have the particularity of being highly immunogenic and should be avoided in immunocompromised patients. However, repeated use of a vaccine with the same vector can become ineffective via a humoral immune response directed against the vector itself [38,43,44].

#### 1.2.6. Personalized Vaccination

The latest strategies discussed above focus on tumor antigens shared by numerous types of cancer and a wide range of patients. However, the immunogenic effects of these vaccines are limited because many of these tumor antigens are recognized as self-antigens and do not trigger an immune response. Secondly, the expression of tumor antigens from different tumor tissues and cancers can be highly variable due to biological tumor heterogeneity. Indeed, studies have shown that most non-synonymous mutations found in a given patient appear to be unique to that specific tumor. Thus, the expression of tumor antigens is subject to great variability among patients [41,45]. Moreover, tumor tissues use several escape mechanisms to evade anti-cancer immunity (e.g., decreased expression by cancer cells of tumor antigens by MHC I, overexpression of IDO, etc.) [30,46,47].

The development of high-throughput sequencing techniques in recent years makes the identification of tumor-specific mutations (also called the mutanome) possible, producing TAAs with high specificity. By comparing the genome of malignant and healthy tissues of a given patient, these techniques avoid interpreting germline variants as neoepitopes. These methods have enabled the development of personalized cancer vaccines [41,45]. 

In humans, only a small portion of the mutated genes will result in the formation of tumor antigens with immunogenic neoepitopes [45,48]. Therefore, it is important to choose mutations that will produce epitopes that are as immunogenic as possible. In an experiment on cancer-bearing mice, researchers injected RNA-encoding antigens representing several mutations of their syngeneic tumor. Most non-synonymous cancer mutations were immunogenic and were recognized by CD4+ helper T lymphocytes, resulting in tumor growth control. This suggested that personalized vaccines can be made using algorithms that select mutations from a patient’s mutanome that have the highest probability of creating widely expressed tumor antigens to induce a CD4+ response [45,49]. Indeed, vaccines that encode mutations which provide MHC I- and MHC II-compatible antigens are known in mice to induce tumor rejection [48]. 

Based on these principles, the first-in-human personalized vaccination phase I trial using RNA-based encoding poly-neoepitopes was used in 16 eligible patients with stage III and IV melanoma [48]. For each patient, the investigators created a vaccine based on two separate strands of synthetic coding RNA for a total of 10 mutations. These non-synonymous mutations were chosen depending on the probability of their affinity with MHC I and II, and their level of expression. The production of RNA took 68 days (range: 49–102 days). All patients received a complete treatment regimen with a maximum of 20 doses. The primary endpoint was safety, and no serious side effects (grade III-IV) were reported. Immune responses were detected for 60% of the expected neo-epitopes, and T-cell responses against at least three mutations developed in each patient. Most of the neo-epitopes triggered CD4+ responses. A smaller part developed a CD8+ response only, and a quarter developed a simultaneous CD4+ and CD8+ response. When comparing melanoma recurrences in all patients before and after vaccination, there was a highly significant decrease in the number of longitudinal cumulative recurrent metastatic events (*p* < 0.0001) after vaccination. Two out of five patients with metastatic disease showed objective responses. Eight patients without radiological lesions were recurrence-free throughout the follow-up period (12 to 23 months). A patient combining his personalized vaccine and a PD-1 inhibitor checkpoint had a CR. The results suggest that this vaccine should be investigated to prevent relapse and as combination therapy with anti-PD-1 immunotherapies [48]. 

## 2. Development of Vaccines for Viral-Induced Head and Neck Cancers

### 2.1. Human Papilloma Virus in HNSCC

The human papillomavirus (HPV) is an oncogenic double-stranded DNA virus. This virus is asymptomatically transmitted through direct skin-to-skin or skin-to-mucosa contact via micro-abrasions in the epithelium infecting the basal cells of the mucosa. The involvement of HPV in the genesis of oropharyngeal cancers (OPCs) was first reported 38 years ago [50]. In recent years, HPV has been officially recognized to have a causal role in OPC with strong epidemiological evidence in the oropharynx [51]. It is responsible for 27% of oropharyngeal cancers in France [52] and about 80% in Sweden and the USA [53].

HPV-positive OPCs are linked in 90% of cases to HPV-16. In comparison, 70% of HPV-positive cervical cancers are related to HPV-16 and HPV-18. Other serotypes have also been recognized as oncogenic (31, 33, 35, 39, 45, 51, 52, 56, 58, 59, 68, 73, and 82) [21]. HPV DNA rapidly expresses different proteins, including E6 and E7 in infected cells, resulting in the degradation of p53 and the retinoblastoma (RB) gene, respectively, leading to uncontrolled DNA synthesis and cell multiplication [21,54].

HPV vaccination with bivalent (HPV16/18), quadrivalent (HPV6/11/16/18), or nonavalent (HPV6/11/16/18/31/33/45/52/58) vaccines has been recommended in the USA since 2006 for women and 2011 for men [55]. These prophylactic vaccines are developed from virus-like particles composed of different viral L1 proteins, which are the main structural proteins of the virus capsid and contain the targeted immunogenic epitopes of the virus. These vaccines allow the production of IgG antibodies directed against the virus to prevent cellular infection. However, these vaccines do not act as therapeutic agents once HPV-induced malignancy is established. This is due to the L1 protein no longer being expressed during the oncogenic phase. As the viral DNA integrates into the cell’s genome, the integration process deletes certain genes, including the L1 protein, rendering the prophylactic vaccine that targets L1 inefficient. Moreover, blocking the virus’ entry into cells is not relevant when it comes to producing a therapeutic effect on already established HPV-dependent cancers. Therapeutic vaccines must therefore target other HPV antigens [21,38,56]. 

Several randomized clinical studies have shown that prophylactic vaccines are more than 90% effective in preventing ano-genital HPV infections and pre-cancerous lesions [55], but data are limited in head and neck cancers. According to a 2015 meta-analysis based on data from 14 high-income countries, after five to nine years of HPV vaccination, cervical intraepithelial neoplasia grade 2+ (CIN2+) decreased significantly by 51% among vaccinated girls aged 15–19 years and by 31% among women aged 20–24 years [57]. In a Cochrane review published in 2018 involving 26 studies and 73,428 participants, it was demonstrated with high evidence that the vaccines decreased rates of CIN-2 from 164/10,000 to 2/10,000, CIN-3 from 70/10,000 to 0/10,000, and adenocarcinoma in situ from 9/10,000 to 0/10,000 in women aged 15–26 years [58].

In the United States, between 2011 and 2014, 18.3% of the population aged 18–33 years had received at least one dose of an HPV prophylactic vaccine before the age of 26 years (29.2% in females versus 6.9% in males). The prevalence of oral HPV infections (16/18/6/11) was significantly reduced between the vaccinated population and the non-vaccinated population (0.11% vs. 1.61%, Padj = 0.008). Moreover, the prevalence of oral HPV infections (16/18/6/11) was also significantly reduced in the vaccinated male population compared to the non-vaccinated male population (0.0% vs. 2.3%, Padj = 0.007) [55]. In Costa Rica, a study comparing the efficacy of the bivalent vaccine versus a control group (randomization 1:1) was conducted in 6466 women aged 18 to 25 years. After four years of follow-up, the efficacy of the vaccine against oral HPV16/18 infections was estimated to be 93.3% [59].

There is hope that preventive HPV vaccination can also reduce the occurrence of HPV-related OPC. In models of canine papillomavirus-associated oral cancer (COPV), vaccination against COPV L1 has been shown to protect against the development of oral carcinoma. Passive transfer of serum immunoglobulins from immunized dogs has also been shown to be protective. Furthermore, antibodies to the HPV-16 L1 protein have been found in the saliva of vaccinated women [60]. To date, no study has been able to significantly demonstrate the preventive effect of HPV vaccines in the oncogenesis of OPC in humans. According to Gillison et al. [61], the effects of prophylactic vaccination targeting HPV on OPCs would not be seen before 2060, assuming high HPV vaccine efficacy and high population coverage.

### 2.2. Development of Therapeutic Vaccines against HPV-Related Antigens for HNSCC

Three prophylactic HPV vaccines directed against the L1 protein of the viral capsid exist but appear to be ineffective in the treatment of HPV-induced OPC. However, HPV-infected cancer cells provide non-host antigens, making these a potential target for vaccine development [62]. The oncoproteins E6 and E7 are ideal exogenous targets because they are constantly necessary and exclusively produced in cancer cells [21,38,62]. Many clinical trials are investigating this rationale with live vector, peptide or protein, nucleic acid, and cell-based vaccines. These strategies have all been extensively tested in ano-genital cancers. For example, an HPV vaccine investigated in genital neoplasia was shown to be efficacious. Kenter et al. [63] developed a long-peptide vaccine targeting E6 that resulted in a clinical response in 79% of patients with HPV-16-positive vulvar grade III intraepithelial neoplasia; 47% of these patients also had a complete response. The use of these treatments in HNSCC remains, however, largely under-investigated despite the increasing incidence of HPV-dependent cancers of the head and neck.

In a phase I study, 16 patients with advanced HNSCC received a “Trojan” peptide vaccine based on a peptide derived from the melanoma antigen E (MAGE)-A3 or HPV-16 E7 antigen. Peptide vaccines were solubilized in Montanide ISA 51 and GM-CSF. This study involved two cohorts of patients, one with a vaccine targeting the MAGE-A3 antigen (*n* = 7) and the second HPV-16 E7 (*n* = 9). These patients all received four doses. Eighty percent of the HPV16 E7 cohort and 67% of the MAGE-A3 cohort had a specific T-cell response with a significant correlation between the presence of specific T-cell responses and humoral responses. Only one patient vaccinated against MAGE-A3 had stable disease (SD) for 10.5 months; all others had PD according to RECIST criteria. Most of the reported side effects were grade I; however, one patient received only one dose and was excluded from the trial due to a severe adverse event (SAE) [64,65].

In another phase I/IIa trial, a DNA-based vaccine directed against the E6/E7 antigens of HPV16 and 18 and encoding interleukin-12 (MEDI0457) was tested in 22 patients with HPV-related OPCs before (cohort 1) and after (cohort 1 and 2) curative treatment [66]. The part of the vaccine targeting HPV was already known to induce a strong immune response in HPV-driven high-grade cervical dysplasia [67]. Elevated antigen-specific T-cells were observed in 18 patients out of 21 evaluable patients by INF-γ ELISpot. All patients showed humoral responses against at least one HPV-specific antigen. For four of the five patients for whom post-vaccination tumor samples were obtained, the CD8/FoxP3 ratio increased. The number of perforin-positive immune infiltrates also increased among all five patients. One patient developed a metastasis, motivating the initiation of checkpoint inhibitor therapy. After four doses of nivolumab, the patient achieved a complete response (CR) which was maintained for more than 18 months [66]. An additional phase I/IIa study in R/M HNSCC combining the MEDI0457 vaccine and durvalumab, a monoclonal antibody targeting PD-L1, is now completed (NCT03162224). Preliminary results, presented in an ESMO abstract, showed an ORR of 22.2% with 3 CR and 3 PR. Tumor-infiltrating CD8+ T cells and peripheral HPV-specific T cells were also increased [68].

A live attenuated vaccine based on Listeria monocytogenes, ADXS11, is currently being tested in phase II studies in the pre-operative period of transoral robotic surgery for oropharyngeal HPV-positive cancers (NCT02002182). This window of opportunity study is particularly interesting for translational research [69]. This vaccine secretes a fragment of the LLO fused to the oncoprotein HPV-16 E7 and has already shown efficacy in mice implanted with a HPV-induced HNSCC [70,71]. So far, five of the eight patients who completed their vaccination regimen developed T-specific lymphocytes in an IFN-γ ELISpot test. The cytokines CCL22 and CXCL10 showed an increasing trend after vaccination with significant correlation between the CCL22 cytokine and the specific T cell response. Multiplex immunofluorescence showed increasing tumor infiltration of CD4+ and CD8+ T cells in post-treatment samples among four patients [72]. Results of the completed study are pending.

Recently presented at the 2021 American Society of Clinical Oncology (ASCO) meeting, a phase II study (NCT04287868) in advanced HPV-positive cancers investigated triplet therapy consisting of: PDS0101 (a liposomal multipeptide therapeutic vaccine targeting HPV16 E6/E7) with M9241 (a tumor-targeting immunocytokine composed of IL-12 heterodimers fused to a monoclonal antibody targeting free DNA in necrotic tumor areas) and M7824 (or bintrafusp alfa, a bifunctional fusion protein targeting TGF-β and PD-L1). Among the fourteen patients included, three had an oropharyngeal cancer, four experienced a grade 3 treatment-related AE, and 10 had an objective response, including one complete response and nine partial responses (including two patients with oropharyngeal cancer). Nine out of ten patients still had an objective clinical response after a median follow-up of five months. Translational investigations are currently ongoing [73].

The first report of a phase I (NCT04180215) study investigating the safety, tolerability, and anti-tumor activity of HB-201 and HB-202, two virus-based vaccines, was also presented at ASCO 2021. HB-201 and HB-202 are replicated live-attenuated vectors based on the lymphocytic choriomeningitis and the Pichinde viruses, respectively, which express an identical E7E6 non-oncogenic fusion protein of HPV16. Phase I is evaluating various schemes and dose levels of HB-201 in monotherapy and HB-201 and HB-202 as alternating 2-vector therapy, administered intravenously (IV) with or without initial intratumoral administration. Twenty-five patients with heavily pre-treated advanced HPV16-positive cancers were recruited and 72% of these have oropharyngeal cancers. Only fatigue has been reported as an SAE along with grade III AEs related to the study drug. Eighteen patients were evaluable for clinical response. Of the sixteen patients receiving HB-201 in monotherapy, two patients developed a PR and six had SD. The two patients who received alternating HB-201 and HB-202 both experienced SD [74].

Another TAA in HPV-related cancers is the p16INK4a protein that is overexpressed in HPV-associated head and neck cancers. The p16 protein has also become a common pathological marker in the diagnosis of HPV-related OPC [54,56]. A phase I/IIa study in 24 patients with advanced HPV-associated cancers, including six HNSCC, investigated an LP vaccine directed against the p16^INK4a^ protein and found that 64% of patients had SD and 36% had PD (median follow-up: 5.6 months). The vaccine was considered safe. Moreover, 14 out of 20 and 5 out of 20 patients showed the presence of specific CD4+ T cells and CD8+ T cells, respectively, while 14 out of 20 patients developed antibodies directed against the targeted protein [75].

To date, many phase I and II studies have investigated the efficacy of vaccine-based therapies in HPV-induced HNSCC (Table 1). However, to the best of our knowledge, there is currently no phase III study evaluating a vaccine that targets HPV-associated antigens in head and neck cancer.

### 2.3. Vaccination in Epstein–Barr Virus-Induced Undifferentiated Nasopharyngeal Carcinoma 

Another type of head and neck cancer is undifferentiated nasopharyngeal carcinoma (NPC), which affects the epithelium of the nasopharynx. NPC is induced by the Epstein–Barr virus (EBV) in over 95% of cases. It is common throughout South-East Asia, where the highest rate of 17.4 per 100,000 is reported in Singapore. The incidence in Europe and North America is lower at 0.5 per 100,000 [77]. EBV is known to be responsible for approximately 200,000 cancers worldwide each year, including several lymphomas such as Burkitt’s lymphoma or Hodgkin’s lymphoma [78]. Unlike other oncoviruses, such as the hepatitis B virus and HPV, there is no approved prophylactic vaccine against EBV. Only a protein vaccine directed against the gp350 protein of the viral envelope has shown to decrease infectious mononucleosis, even though it failed to influence the rate of EBV acquisition [79]. Another study showed, however, that elevated titers of EBV-neutralizing antibody and anti-gp350 antibody were indicative of low-risk biomarkers for NPC [80]. Further studies are needed to draw conclusions. Other prophylactic vaccines are still under development [81].

The current locoregional treatment for locally advanced NPC includes neoadjuvant chemotherapy followed by chemoradiation [82], but in advanced cancers, this treatment fails in 25% of cases. In addition, these cancers are known to have a high metastasizing capacity [83]. Most of these carcinomas express two proteins: the Epstein–Barr nuclear antigen 1 (EBNA1) and the latent membrane protein 2 (LMP2). The first is a protein involved in the maintenance of the episomal DNA virus and has several epitopes stimulating a CD4+ response. The second is a membrane protein involved in cell growth activity in epithelia and has many epitopes stimulating a CD8+ response [78,84]. A first phase I study published in 2002 evaluated the effects of injecting a vaccine based on autologous monocyte-derived DCs into the inguinal lymph nodes of 16 patients with local recurrence or metastatic NPC after conventional therapy. The autologous cells collected were stimulated with HLA-A1101-, A2402-, or B40011-restricted epitope peptides from the LMP-2 protein. Nine of the twelve patients who received HLA-A1101 or HLA-2402 specific peptides demonstrated an increase in the production of INF-γ by T cells for at least three months, as assessed by an INF-γ ELISpot test. Two patients vaccinated with the HLA-A1101-restricted peptide had a partial response (PR) [85]. 

In a Chinese study, 16 HLA-A2 patients with stage II-III NPC were vaccinated after (chemo)radiotherapy with autologous DCs stimulated by a restricted HLA-A2 LMP2A peptide. After vaccination, serum levels of interferon gamma and interleukin-2 were significantly increased with an elevation in the percentage of CD4+ and natural killer cells; however, serum EBV DNA levels were significantly decreased among the nine patients who showed a skin response to the LMP2A peptide in a delayed peptide hypersensitivity test. The vaccine was well tolerated and no recurrence was detected during follow-up (median follow-up 6.87 months) [86]. 

A phase I study published in 2014 using a vaccine based on the modified vaccinia Ankara virus (MVA) was tested on 16 patients after standard treatment. This virus encodes an inactive form of the LMP2 protein and the immunogenic C-terminal half of the EBNA1 protein. Vaccination was well tolerated, and 8 out of 14 patients showed an immune response to at least one antigen detected by INF-γ ELISpot [84]. A similar study tested the same vaccine in 18 Chinese patients after standard treatment. Fifteen of the eighteen patients experienced an increase in specific T cells in an ELISpot test after vaccination [87]. Based on the available data from these two studies, 18 and 12 patients exhibited an increased T-specific T cell response to the EBNA1 and LMP2 antigens, respectively, following vaccination [78]. A phase II clinical trial is currently underway to evaluate the clinical efficacy of this vaccine as a primary endpoint (NCT01094405).

A phase II study evaluated the efficacy and safety of a vaccine based on autologous DCs infected ex vivo with a recombinant Ad5f35 virus encoding for an inactive LMP1 protein and for full-length LMP2, and then matured with a cytokine cocktail. This vaccine was administered to 16 patients with metastatic nasopharyngeal EBV-induced carcinoma. No significant toxicity was identified. No increase in LMP1/LMP2-specific T cells was demonstrated in ELISpot tests after vaccination. However, a delayed type of hypersensitivity response was observed in 9 out of 12 patients post-vaccination. One patient developed a PR and two patients had SD [88].

Clinical trials investigating other therapeutic vaccines are ongoing (Table 2).

## 3. Therapeutic Vaccines Targeting Non-Viral Antigens in HNSCC

HPV-negative HNSCC is mainly related to chronic alcohol and tobacco intoxication, explaining why the mutation spectrum of the genes involved in oncogenesis is significantly different compared to HPV-induced cancers [89]. The gene most frequently altered in HPV-positive cancers is *PIK3CA*. In HPV-negative HNSCC, the most frequently altered genes code for *TP53* and *CDKN2A/B,* which are implicated in the DNA repair p53 and cell cycle pathways [90].

Targeting p53 is a valuable choice in HPV-negative head and neck cancers, as opposed to HPV-related cancers where the protein is unexpressed. Indeed, HPV-negative HNSCC is characterized by a high mutation rate of the *TP53* gene that leads to an accumulation of the p53 protein. Accumulation of this protein is associated with enhanced presentation of a wild-type sequence of p53 peptides to immune cells. Wild-type sequence p53 epitopes have been identified as inducing CD8+ cytotoxic T cells when other epitopes stimulate CD4+ T cells. Thus, a vaccine based on p53 wild-type peptides has been considered [91,92]. A phase I clinical trial published in 2014 [93] used autologous DCs stimulated ex vivo by p53 peptides in 16 HLA-A2.1+ patients with treated HNSCC. No grade II-IV adverse events were reported. Eleven patients (69%) showed a positive response to tetramer analysis. Of these, four showed a positive ELISpot test. In addition, CD4 + CD25 + CD39+ regulatory T cells (T-reg) count in flow cytometry was significantly lower after vaccination. Disease-free survival at two and three years was 88% and 80%, respectively. These values are increased compared to a study conducted in 2010 in the same institution examining the effect of radiochemotherapy in patients with advanced HNSCC. Eight patients were considered p53-positive by immunohistochemistry, and eight patients were considered p53-negative. No statistical difference in disease-free survival was found between both groups [93].

Survivin-2B is a member of the inhibitor of apoptosis protein family and is overexpressed in most malignancies. In a phase I study of 10 HLA-A24-positive patients with advanced or recurrent oral cancer, a peptide vaccine targeting survivin-2B was found to be safe. An increase in the level of T cells specific to survivin-2B was demonstrated by tetramer analysis in six patients. One patient had a PR and the nine other patients had PD [94]. 

In another clinical trial, the Allovectin-7 vaccine, a DNA plasmid-lipid complex coding for the HLA-B7 heavy chain and β-2 microglobulin, was injected intratumorally into 69 patients with incurable HNSCC in a phase I-II study that also included 60 HLA-B27-negative patients. The goal of treatment was to increase the presentation of HLA-B7 on tumor cells and thus promote an enhanced anti-cancer immune response. The treatment was well tolerated without grade III or IV side effects. Of the 69 patients, 33% had a clinical response after the first vaccination cycle (10% PR and 23% SD). After the second cycle, one patient had a CR. The authors concluded that there were no obvious predictors of success with the use of Allovectin-7 in this cohort [95]. This vaccine has since been tested in phase III trials in melanoma but did not meet the study objectives of tumor response and increased overall survival compared to chemotherapy. A phase II/III trial was initiated in head and neck cancers but was cancelled without any results being published [96].

Investigators presented the first results from a phase I trial testing the concomitant use of tadalafil and injection of autologous pulsed DCs with the Mucin-1 (MUC1) peptide in patients with surgically eligible, recurrent or second primary HNSCC [97]. Tadalafil had already demonstrated immune effects in HNSCC, such as a decrease in myeloid-derived suppressor cells, T-regs with increased CTL, and CD4+ in tumor tissue [98]. On the other hand, MUC-1 is overexpressed and under-glycosylated in HNSCC compared to normal tissues, making it an ideal target. The combined treatment was well tolerated without any serious adverse events (SAEs). Only two patients developed antibodies against MUC1. However, the concomitant use of the vaccine and tadalafil resulted in a decrease in PD-L1+ macrophages at the tumor edges, an intra-tumoral decrease in FoxP3+ T-regs, and an increase in intra-tumoral CD8+ CTLs associated with an augmentation of the CD69 early activation marker [97].

Another target entity among TAAs is germline antigens. In HLA-A24-positive patients with advanced HNSCC, a phase II clinical study investigated a vaccine based on short peptides derived from germline antigens, lymphocyte antigen 6 complex locus K (LY6K), cell division cycle associated gene 1 (CDCA1), and insulin-like growth factor-II mRNA-binding protein 3 (IMP3). The primary endpoint was OS, which was significantly increased in the HLA-A24-positive group (*n* = 37) compared to the HLA-A24-negative control group (*n* = 18), and median survival was 4.9 vs. 3.5 months, respectively (*p* < 0.05). However, there was no significant difference in PFS. One patient had a CR over 37 months and nine others had SD for three months according to RECIST criteria. In the vaccinated group, specific T lymphocytes were demonstrated for the LY6K-, CDCA1-, and IMP3 peptides after vaccination, observed in 85.7%, 64.3%, and 42.9% of patients, respectively. Patients who developed CTLs for multiple antigens demonstrated better clinical responses. Indeed, the group of patients with specific T cells against three peptides had a median survival of 19.5 months [99].

Currently, cell-based, personalized vaccination is being evaluated in a phase II study in patients with advanced HNSCC (*NCT02999646)*. The vaccine, MVX-ONCO-1, is a personalized vaccine made from irradiated, genetically engineered, and dead tumor cells from the patient that are then injected using cell encapsulation technology. This technology enables the sustained release of GM-CSF, with dead tumor cells releasing tumor-antigens. In a previous phase I study conducted in 15 patients with solid tumors, no serious side effects were reported and more than 50% of the patients had a PR or SD [100]. Preliminary results of 11 patients, from two different clinical trials (NCT02193503 and NCT02999646), with advanced or metastatic HNSCC, relapsing after at least one line of systemic therapy, have recently been published. All patients received at least five administrations of MVX-ONCO-1 over eight weeks. The treatment was considered as safe and tolerable. Four patients had SD, two patients had a PR, and two patients had a CR. The two CRs were long-lasting, with both patients able to cease anticancer treatment for six and 24 months [101].

Another example, the AlloVax vaccine, is based on chaperone protein-enriched tumor cell lysate from the patient’s tumor (calreticulin, hsp70, hsp90, and gr94/gp96 as sources of tumor neoantigen) and the AlloStim adjuvant, which is composed of ex vivo differentiated Th1 memory T cells expressing CD40L and INF-γ. Ten patients with advanced chemo-resistant HNSCC were recruited to a phase II study. The vaccine was well tolerated. Fifty percent of the patients showed a visible clinical response, which correlated with decreased cytotoxic T-lymphocyte-associated protein (CTLA)-4 expression and increased CD3+ infiltrating T cells in tumors [96,102].

Vaccines targeting other tumor antigens are being investigated in clinical trials (Table 3).

## 4. Combining Immune Therapies

As previously explained, vaccine-based therapies are rarely effective as monotherapy, suggesting that the immunosuppressive environment of the tumor controls vaccine-activated T cells. Immune tumor escape may be the result of several mechanisms: activation of inhibitors of the immune checkpoint through PD-1/PD-L1; CTLA4; T cell immunoglobulin- and mucin-domain-containing molecule (TIM)3; lymphocyte activation gene (LAG)3; extrinsic pathways mediated by T-regs or myeloid-derived suppressor cells; and the secretion of cytokines such as TGF-β. Thus, there are good reasons to support the combination of several immunotherapy approaches [103]. For example, combining vaccines with PD-(L)1 inhibitors may induce a tumor T cell-specific response and block the immunosuppression induced by the PD-1/PD-L1 pathway [76].

A phase II study [76] investigated the efficacy of combining a long peptide vaccine, ISA 101, with nivolumab in 24 patients with incurable HPV-induced cancers, including 22 OPC. ISA 101 targets the HPV-16 oncoproteins E6 and E7 and had previously demonstrated efficacy in a cervical cancer study [63]. Patients were treated with three doses of the ISA 101 vaccine on days 1, 22, and 50, and received nivolumab 3 mg/kg intravenously on day 8 and then every 2 weeks thereafter. The primary objective was the overall response rate (ORR), as measured by RECIST v1.1 criteria. Among the 24 patients, eight had a clinical response (2 CR and 6 PR) for an ORR of 33% (90% CI, 19–50%). Responses were durable in five of the eight patients (63%). Overall survival (OS) at 12 months and median OS were 70% and 17.5 months, respectively. Two patients experienced grade III and grade IV SAEs requiring discontinuation of treatment. The immune response according to INF-γ ELISpot was not correlated with any efficacy end points. These first results demonstrated an improved ORR and 12-month OS compared to the reference studies investigating anti-PD1 in monotherapy (nivolumab in CheckMate 141 [104] and pembrolizumab in Keynote-012 [105,106] and Keynote-055 [107]). However, the therapeutic efficacy of the ISA 101 vaccine in combination with nivolumab in HPV-induced HNSCC needs to be investigated in larger randomized studies before any real conclusions can be drawn [76]. 

An additional phase I clinical trial [108] combined a MVA virus-based vaccine expressing wild-type p53 transgene with pembrolizumab in 11 patients with advanced solid cancers, including one patient with HNSCC. Three patients, including the patient with HNSCC, had SD, and two of the three also experienced a specific CD8+ T cell increase. One patient experienced a grade V side effect (myocarditis) that was possibly attributed to the vaccine [108].

Another interesting ongoing phase I/II study (NCT02955290) is investigating the efficacy of CIMAvax vaccine in HNSCC and non-small cell lung cancer (NSCLC) in combination with nivolumab or pembrolizumab. The vaccine contains a chemical conjugate of recombinant human EGF with the P64k protein derived from Neisseria meningitides. The aim of this vaccine is to create antibodies against EGF, thus preventing the ligand from binding to its receptor, which is overexpressed on the surface of neoplastic cells. Already extensively investigated in NSCLC, this vaccine was evaluated in a phase III study in patients with advanced NSCLC previously treated with chemotherapy. A significant increase in median overall survival compared to the control group (12.43 months and 9.43 months, respectively) was observed, especially if the baseline EGF concentration was high (14.66 months) [109,110].

## 5. Challenges and Future Perspectives of Vaccines in HNSCC 

Two of the main challenges for vaccines, especially personalized vaccines, are manufacturing costs and patient access [62]. High-throughput sequencing and bioinformatics can now produce personalized RNA- or cell-based personalized vaccines. However, these vaccines are currently being studied in trials and not readily available for patients with advanced HNSCC for whom rapid and efficient therapeutic care is vital. Despite the documented efficacy of vaccines, production time and the time to achieve a therapeutic effect may be too long for some patients, potentially limiting their use. In addition, the efficacy of CTLs produced by vaccines may be rendered ineffective due to the immunosuppressive tumor microenvironment. Studying these vaccines in combination with ICIs to enhance efficacy via synergistic effects is therefore a highly promising approach [62].

Another promising alternative approach under investigation is the use of an autologous blood cell-based vaccine that is engineered using red blood cells (RBCs). RBCs are made from erythroid precursors that are genetically modified to express proteins of interest (e.g., MHC type I with a peptide, co-stimulatory molecule, interleukin) on their surface. After expression of the proteins of interest on their surface, the engineered cell is enucleated, which allows the cells to be injected into the patient without genetic modifications induced ex vivo. Moreover, as RBCs are confined to the blood vessels and spleen, this technique avoids the side effects found with other immunomodulatory proteins that diffuse to all organs. Pre-clinical studies investigating RTX-240, which is a genetically engineered red blood cell expressing 4-1BBL and IL-15/IL-15Rα fusion, have shown that the use of this novel technique resulted in T cell and NK cell expansion, memory T cell formation, and tumor growth control in a B16-F10 melanoma model [111]. The investigated product was shown to be safe and well tolerated, allowing the initiation of a phase I-II study of RTX-240 monotherapy and in combination with pembrolizumab in advanced solid tumors (NCT04372706). 

Currently, a phase I clinical study is investigating the safety of RTX-321 monotherapy as the primary endpoint in HLA-A*02:01-positive patients with advanced HPV-positive cancers (NCT04672980). RTX-321 is a drug composed of red blood cells that are transformed into artificial antigen-presenting cells that harbor on their surface the HLA-A*02:01 presenting an HPV E7 peptide, the co-stimulatory molecule 4-1BBL and IL-12. In in vitro assays, RTX-321 has shown activation of HPV-specific T cells with effector function [112].

The use of vaccines in the neoadjuvant setting is also worthy of investigation. Several studies in mice have shown promising outcomes when a therapeutic vaccine was administered before surgery [113,114]. For example, in an experiment involving 22 laboratory mice (inbred strain C57BL/6) carrying B16F10 melanomas, a vaccine based on two peptides was administered nine days prior to surgery. After surgery, 21 of the 22 mice were free of tumor recurrence. Following neoadjuvant vaccination, the frequency of CD8+ T cells and of CD4+ T cells multiplied by more than 15 and four, respectively, compared to unvaccinated mice [113]. In a phase II study, three doses of the sipuleucel-T vaccine were administered in the six weeks prior to surgery as neoadjuvant therapy to patients with localized prostate cancer. A significant increase in CD3+, CD4+FoxP3- and CD8+ was demonstrated by comparing pre- and post-treatment samples from the treated group against the control group. Six patients (16%) had a lower post-treatment Gleason score compared to their baseline biopsies; however, in eight patients (22%), the Gleason score increased. No information on patient survival was provided [115]. 

Currently, several clinical trials are underway to investigate the value of therapeutic vaccines in solid tumors, either alone or in combination neo-adjuvant regimens. Studies investigating new treatments using window of opportunity study designs could provide valuable information. These studies enable the investigation of new molecules for biological effectiveness in primary cancers without delaying standard curative treatment. Window of opportunity studies also provide data on molecular and clinical activity as well as possible predictive biomarkers [69]. However, the challenge of this study design, particularly in vaccine trials, is whether an immunological response could develop from a short vaccination regimen. In our institution, an umbrella study based on a window study design has recently started. The first arm of the study will investigate the effect of a long peptide vaccine targeting the IDO enzyme versus a control group (NCT04445064).

Another area of emerging research aims to increase the immunogenicity of therapeutic vaccines. One technique under investigation is photochemical internalization (PCI). This technique uses the concomitant administration of a vaccine and a photosensitizer. The photosensitizer, once illuminated, lyses the membrane of endosomal vesicles by secreting very locally reactive oxygen species and releasing the contents into the cell’s cytosol (such as a DNA, RNA, or peptide vaccine). In in vitro experiments, the concomitant use of PCI with a short peptide increases the number of MHC I-peptide complexes on the surface of DCs. In addition, stimulation of CD8+ antigen-specific T cell proliferation was 30- to 100-fold more effective with the concomitant use of PCI and the peptide vaccine versus the peptide alone. Furthermore, the authors demonstrated the presence of a strong increase (greater than 10-fold increase in the median percentage) in HPV-directed CTLs in mice vaccinated with PCI and the HPV16 E7 peptide compared to mice vaccinated with the HPV16 E7 peptide alone [116]. In a first phase I study (NCT02947854) [117], the fimaVacc vaccine, combining PCI therapy, a TLR agonist, and an HPV16 E7 peptide, was well tolerated in healthy participants. ELISpot and flow cytometry demonstrated the presence of CD4+- and CD8+-specific HPV T cells following vaccination.

## 6. Conclusions

Although therapeutic vaccines appear to be safe, the main challenge is how to achieve an effective and long-lasting specific immune response against tumor antigens. Although the clinical response from therapeutic vaccines alone appears to be poor, the results of treatment combinations, particularly with immune checkpoint inhibitors, are encouraging. Larger-scale studies need to be conducted to clarify their real therapeutic potential, but the difficulty here lies in patient selection. Most studies involving therapeutic vaccines have been carried out in patients with advanced and/or recurrent cancers; that is, a heavily pre-treated population. This factor may underestimate the true clinical efficacy of these novel immunotherapies.

## Figures and Tables

**Figure 1 cancers-13-06041-f001:**
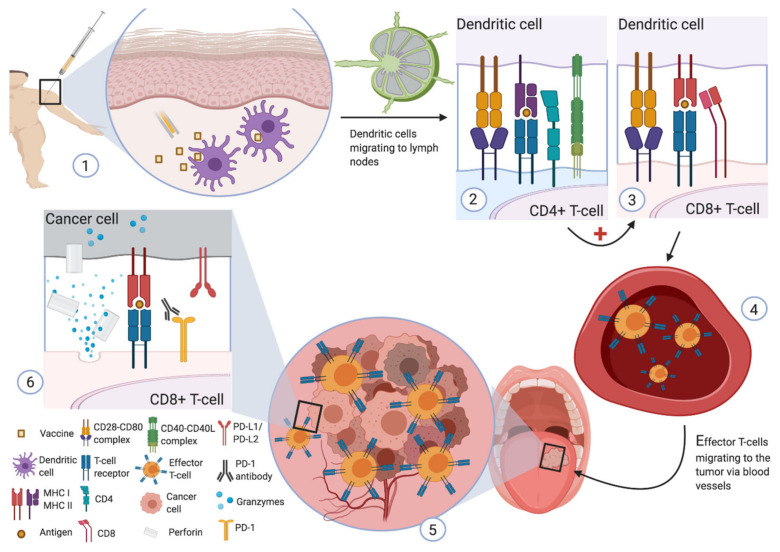
Schematic representation of therapeutic vaccination using a peptide vaccine in a patient with tongue cancer. (**1**) Injection of the vaccine subcutaneously containing one or more peptides associated with tumor cells. Vaccine compounds are phagocyted by locally present dendritic cells. After migration of dendritic cells into lymph nodes, stimulation of CD4+ (**2**) and CD8+ (**3**) T lymphocytes by recognition of the antigens (formed from the vaccine) through interaction between the T cell receptor and the corresponding major histocompatibility complex. CD4+ effector T-lymphocytes produce surface proteins and cytokines for the activation of B lymphocytes, dendritic cells, and macrophages. They are also involved in the production of memory T cells. (**4**) Then, effector T-lymphocytes migrate through the blood vessels to the tumor. (**5**) Increased lymphocyte infiltration and inflammation within the cancerous tissue. (**6**) Tumor antigen recognition by cytotoxic CD8+ T lymphocytes via major histocompatibility complex type I. Activation of cytolytic compounds (as granzymes and perforin) to kill the cancer cells. Note the probable synergistic effect of immune checkpoint inhibitors and therapeutic vaccines.

**Table 1 cancers-13-06041-t001:** Clinical trials in vaccines targeting HPV-associated antigens.

Identifier and Reference	Vaccine ± Other Therapy	Phase	Type of Vaccine	N	Target Antigens	Population	Primary Endpoint	Status
NCT00257738 Ref. [65]	GL-0810/GL-0817	I	Peptide	16	MAGE-A3 * (*n* = 7) /HPV16-E7 (*n* = 9)	HPV16-positive or MAGE-A3-positive R/M-HNSCC	Safety	Completed
NCT02163057 Ref. [66]	MEDI0457	I/IIa	DNA	22	HPV16 E6/E7 HPV18 E6/E7	Advanced HPV-related HNSCC	Safety	Completed
NCT03162224 NA	MEDI0457 + Durvalumab	I/IIa	DNA	±35	HPV16 E6/E7 HPV18 E6/E7	R/M HNSCC HPV+	Safety and efficacy	Completed
NCT02002182 NA	ADXS11	II	Live (Listeria Monocytogenes)	±15	HPV16 E7	Surgically elected HPV+ oropharyngeal SCC	Change in specific CD8+ CTL response and safety	Active, not recruiting
NCT02291055 NA	Durvalumab ± ADXS11	I/II	Live (Listeria Monocytogenes)	±66	HPV16 E7	R/M HPV16+ HNSCC or cervical cancer	I: Safety II: PFS and safety	Active, not recruiting
NCT03418480 NA	HARE-40	I/II	RNA	±44	HPV16 E6/E7	I: Advanced HPV16+ HNSCC II: Advanced HPV16+ cancer (HNSCC, anogenital, penile, cervical)	I: Safety II: Efficacy and significant increase in specific immune cells	Active, not recruiting
NCT02426892 Ref. [76]	ISA101 + Nivolumab	II	Peptide	24	HPV16 E6/E7	Incurable HPV16+ cancers (22 oropharyngeal cancers, 1 anal cancer, and 1 cervical cancer)	Efficacy	Active, not recruiting
NCT03258008 NA	ISA101b + Utomilumab	II	Peptide	±27	HPV16 E6/E7	HPV16+ incurable oropharyngeal cancer	Efficacy	Active, not recruiting
NCT02865135 NA	DPX-E7	Ib/II	Peptide	±11	HPV16 E7	Positive HLA-A*02 patients with HPV-related head and neck, cervical or anal cancer.	Safety	Active, not recruiting
NCT03260023 NA	TG4001 + Avelumab	Ib/II	Live (modified vaccinia Ankara virus)	±52	HPV16 E6/E7	HPV-related carcinomas	I: Safety II: Efficacy	Recruiting
NCT02526316 NA	p16 vaccine + concurrent cisplatin-based chemotherapy	I	Peptide	±11	p16	p16-positive cervical, vulvar, vaginal, penile, anal, or head and neck cancer	Immune response	Completed
NCT01462838 Ref. [75]	p16 vaccine	I/IIa	Peptide	24	p16	HPV-associated cancers (including 6 HNSCC)	Immune response	Completed
NCT04260126 NA	PDS0101 + Pembrolizumab	II	Peptide	±96	HPV16 E6/E7	HPV16+ R/M HNSCC and HPV-related esophageal SCC	Efficacy	Recruiting
NCT04369937 NA	ISA101b + Pembrolizumab + Cisplatin + radiotherapy	II	Peptide	±50	HPV16 E6/E7	“Intermediate risk” HPV-16 associated HNSCC	Efficacy	Recruiting
NCT04534205 NA	BNT113 + Pembrolizumab vs. Pembrolizumab alone	II	RNA	285	HPV16 E6/E7	HPV16 + and PD-L1+ R/M HNSCC	Part A: Safety Part B: Efficacy	Recruiting
NCT04287868 Ref. [73]	PDS0101 + M9241 + M7824	I/II	Peptide	21	HPV16 E6/E7	Advanced HPV16-positive cancers	Objective Response Rate	Suspended
NCT04180215 Ref. [74]	HB-201 ± HB-202	I/II	Virus	±200	HPV16 E6/E7	HPV16-positive cancers	I: dose-limiting toxicities II: ORR	Recruiting
NCT04672980 NA	RTX-231	I	Allogenic aAPC	±63	HPV16 E7	Advanced HPV16 positive cancers	I: safety	Recruiting

aAPC: artificial antigen-presenting cells; HPV: human papillomavirus; R/M: recurrent or metastatic; NA: not available; PFS: progression-free survival; HNSCC: head and neck squamous cell carcinoma. * MAGE-A3 is not a specific HPV-associated antigen.

**Table 2 cancers-13-06041-t002:** Clinical trials in vaccines targeting EBV-associated antigens (non-exhaustive list).

Identifier and Reference	Vaccine ± Other Therapy	Phase	Type of Vaccine	N	Target Antigens	Population	Primary Endpoint	Status
NCT01147991 Ref. [84]	MVA-EL	I	Live (MVA virus)	16	EBNA1 LMP2	EBV-induced NPC in CR after first-line treatment	Safety and IR	Completed
NCT01256853 Ref. [87]	MVA-EL	I	Live (MVA virus)	18	EBNA1 LMP2	EBV-induced NPC in CR or unconfirmed CR	Safety	Completed
NCT01800071 NA	MVA-EBNA1/LMP2	Ib	Live (MVA virus)	22	EBNA1 LMP2	EBV-induced NPC in remission or with current disease for whom no standard therapy is currently appropriate or required	IR and Safety	Completed
NCT01094405 NA	MVA EBNA1/LMP2 vaccine	II	Live (MVA virus)	25	EBNA1 LMP2	Persistent, recurrent, or metastatic NPC that have residual EBV DNA following completion of conventional therapy	Efficacy	Completed
NA Ref. [85]	DC vaccine	I	Autologous DCs	16	LMP2	Local recurrence or metastasis NPC	Safety	Completed
NA Ref. [86]	DC vaccine	?	Autologous DCs	16	LMP2	Stage II-III NPC	IR	Completed
NA Ref. [88]	Ad-ΔLMP1-LMP2 DC vaccine	II	Autologous DCs transducted with an adenovirus	16	LMP1 LMP2	Refractory metastatic NPC	Efficacy	NA
NCT00078494 NA	LMP-2:340–349 or LMP-2:419–427	I/II	Peptide	99	LMP2	Locally controlled anaplastic NPC	IR	Completed
NCT00589186 NA	Ad5F35-LMP1/LMP2-transduced autologous DCs + Celecoxib	II	Autologous DCs transducted with an adenovirus	±35	LMP1 LMP2	Metastatic NPC	Efficacy	Unknown

CR: complete response; DC: dendritic cell; EBV: Epstein–Barr virus; IL: interleukin; IR: immune response; MVA: modified vaccinia Ankara; NA: not available; NPC: nasopharyngeal carcinoma.

**Table 3 cancers-13-06041-t003:** Clinical trials in vaccines targeting non-viral antigens.

Identifier and Reference	Vaccine ± Other Therapy	Phase	Type of Vaccine	N	Target Antigens	Population	Primary Endpoint	Status
NCT00404339 Ref. [93]	Peptide pulsed DCs	I	DC	16	p53	HLA-A2.1-positive patients with treated HNSCC	Safety	Completed
UMIN000000976 Ref. [94]	Survivin-2B vaccine	I	Peptide	10	Survivin-2B	10 HLA-A24-positive patients with advanced or recurrent oral cancer	Safety	Completed
NCT00050388 Ref. [95]	Allovectin-7	I/II	DNA	69	Restore HLA-B7/β2	Persistent or recurrent HNSCC after (chemo)radiotherapy	Safety	Completed
NA Ref. [99]	Peptide vaccine	II	Peptide	55	LY6K, CDCA1 and IMP3	HLA-A24-positive patients with advanced HNSCC	Overall survival	Completed
NCT02999646 [101]	MVX-ONCO-1	II	Personalized	±41	Autologous tumor cells	Advanced HNSCC	Overall survival	Recruiting
NCT01998542 Ref. [102]	AlloVax	II	Personalized	10	Chaperone-enriched tumor cell lysate	Advanced chemo-resistant HNSCC	Efficacy	Completed
NCT03946358 NA	UCPVax	II	Peptide	±47	Telomerase	HPV+ cancers (head and neck, anal, and cervical cancers)	Efficacy	Recruiting
NCT03552718 NA	YE-NEO-001	I	Personalized	±16	NA	Solid cancers in curative post-treatment surveillance period	Safety	Unknown
NCT03548467 NA	CB10.NEO + Bempegaldek-leukin	I/II	Personalized	±65	NA	Locally advanced or metastatic solid tumors	Safety	Recruiting
NCT00019331 NA	Ras vaccine	II	Peptide	NA	Ras	Metastatic solid tumors	IR, Efficacy, and Safety	Completed
NCT00021424 NA	TRICOM vaccine	I	Live (Fowlpox virus)	Max 20	Expression of B7-1, ICAM-1, and LFA-3	Advanced SCC of the oral cavity or oropharynx or nodal or dermal metastases	DLT	Completed
NCT02544880 Ref. [97]	MUC1 vaccine + Tadalafil	I/II	Peptide	16	MUC1	Resectable and recurrent or second primary HNSCC	I: Safety II: IR	Completed
NCT04247282 NA	M7823 ± TriAd vaccine * ± N-803	I/II	Live (Adenovirus)	±40	Brachyury, Mucin-1, and CEA	p16-negative resectable HNSCC	Efficacy	Suspended
NCT04266730 NA	PANDA-VAC	I	Personalized peptide	±6	NA	Advanced lung cancers or HNSCC under Pembrolizumab	Safety	Not yet recruiting
NCT03689192 NA	ARG1 vaccine	I	Peptide	±10	Arginase-1	Metastatic solid tumors	Safety	Recruiting
NCT03311334 NA	DSP-7888 + ICI	Ib/II	Peptide	±84	WT1	Advanced solid tumors	I: Safety II: Efficacy	Recruiting
NCT04445064 NA	IO102	II	Peptide	18	IDO	Curable HNSCC	Biological activity	Recruiting
NCT04470024 NA	DPV-001 + delayed anti-PD1 ± anti-GITR	Ib	Autophagosome-enriched vaccine	56	NA	R/M HNSCC	Safety	Recruiting
NCT05075122 NA	UV1 vaccine + Pembrolizumab + Sargramostism	II	Peptide	75	Telomerase	R/M HNSCC with CPS ≥1	Efficacy	Recruiting

DLT: dose-limiting toxicity; HSV: herpes simplex virus; ICI: immune checkpoint inhibitor; IDO: indoleamine-2,3-dioxygenase; IR: immune response; NA: not available; HNSCC: head and neck squamous cell carcinoma. * TriAd vaccine: ETBX-051 (adenoviral brachyury vaccine) + ETBX-061 (adenoviral Mucin-1 (MUC1) vaccine) + ETBX-011 (adenoviral carcinoembryonic antigen (CEA) vaccine).

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
