# Peer review of "Vaccine-Based Immunotherapy for Head and Neck Cancers"

_cancers, 2021, doi:10.3390/cancers13236041_

Round 1

Reviewer 1 Report

I believe this is a comprehensive review covering a wide range of vaccine modalities relevant to HNSCC.

I have a few suggestions for the authors.

I suggest that the article provides potential mechanism and preclinical evidence of epitope spread seen in therapeutic vaccines to generate antitumor immune response beyond the target antigens.

I suggest that the author to provide their opinion on the low efficacy of DC-based vaccine and some modalities over others in terms of mechanism of antigen-presentation.

I don’t think oncolytic virus (e.g T-VEC) should be considered a therapeutic vaccine as it is not designed to deliver specific tumor antigen but rather generate non-specific innate immune response.

I suggest that the introduction of the institution study (NCT04445064) should be removed from the conclusion as it seems appropriate to be in the conclusion. This could be in the main text.

Please consider a novel approach of autologous blood cell-based vaccine (PBMC or RBC) of direct cytosolic introduction of target epitope peptide (e.g. Rubius, SQZ)

Please consider discussing the data of NCT04287868 liposomal HPV16 E6/7multipeptide vaccine and NCT04180215 virus vector based HPV16 E6/7 vaccine studies presented at ASCO 2021

Reviewer 2 Report

This is a review on vaccine based immunotherapy for head and neck cancers.

MAJOR COMMENTS

The manuscript is well written and easy to read, covering the subject very well.

On line 292-293 it is said that "This virus is asymptomatically sexually transmitted,..." even if  sexual transmission is the most documented way pof spreading, there are also other non-sexual ways, such as skin contact (non sexual), mother to child etc, and that should also be mentioned.

Reviewer 3 Report

This is an extensive review investigating the vaccine-based immunotherapy treatments in head and neck squamous cell carcinoma (HNSCC). This manuscript covers different types of treatments tested in clinical trials, including prophylactic and therapeutic vaccines targeting viral or non-viral antigens.

The conclusion of the authors highlighted the main challenge of this new approach, namely the need to achieve an effective and long-lasting specific immune response against tumor antigens.

Overall, I think the paper is very interesting; I suggest only minor revisions to improve the paper:

Introduction:

- The authors mention the prognostic role of “cold” and “hot” tumor microenvironment in HNSCC. As the importance of the topic, I suggest discussing some studies that have been carried out with the aim to use the immune phenotype as a new prognostic tool for HNSCC (for your convenience, DOI: 10.1002/cam4.3440).

Minor:

- Use the acronym HNSCC instead of SCCHN, since is more commonly used in literature.

- Only minor language corrections should be necessary.
